# The association of types, intensities and frequencies of physical activity with primary infertility among females in Gaza Strip, Palestine: A case-control study

**Amal Dhair** [ID]*, **Yehia Abed**

Faculty of Public Health, Al-Quds University, Gaza Strip, Palestine

* Amaldhair@yahoo.com

## Abstract

### Introduction

Physical activity and energy state balance have fundamentally been related to reproductive system and health. This study explored the relationship between different types, intensities and frequencies of physical activity with primary infertility among women in Gaza Strip, Palestine.

### Methods

A case-control study was conducted in Gaza Strip with the participation of 320 married couples. 160 infertile couples were chosen from five fertility centers registries from 2016 to 2018 and matched residentially with 160 fertile couples. Cases were selected through systematic stratified sampling of five lists categorized according to residency and the determined percentage selected from each list was proportional. Data were collected through a self-administered questionnaire extended by the short form of international physical activity questionnaire and analyzed through SPSS program version 22 by using descriptive analysis, independent T-test, cross-tabulation, and binary logistic regression.

### Results

Low frequency, intensity and duration of physical activity were associated with 3.1 risk of primary infertility (95% CI, 1.60–5.99, $P < 0.001$). Adjustment for age, marital age, age of menarche, refugee status and monthly income provided 3.2 risk (95% CI, 1.55–6.60, $P = 0.002$). Women spending more than 300 minutes a day sedentarily were 2.3 times more likely to have fertility problems than physically active females. Measuring energy expenditure in MET-min/w (Metabolic Equivalent) showed vigorous MET-min/w as negatively associated with the infertility status of females (Interquartile range IQR: 480 for cases and 720 for controls, $P = 0.010$). On the basis of energy expended in kilocalories in relation to weight, results showed the same association (IQR: 564 for case and 864 for controls, $P = 0.011$). No associations were found between moderate activity levels and primary infertility.

**Data Availability Statement:** All relevant data are within the manuscript and its Supporting Information files. All data files are also available

from the 4TU.Researchdata database (doi:10.4121/uuid:5e5e37ce-c66a-44ab-8ee9-715653c6f02d).

**Funding:** The authors received no specific funding for this work.

**Competing interests:** The authors have declared that no competing interests exist.

## Conclusion

Low levels of physical activity and sedentary lifestyle endanger the fertility status of females in Gaza Strip. This may offer the need for endorsing and formalizing adequate physical activity education and awareness protocols in the national reproductive health guidelines and empowering environmental capacity building to alter physical activity-related cultural norms.

## Introduction

Primary infertility is considered when there is "inability to conceive despite cohabitation and exposure to the risk of pregnancy for a period of 12 months or more in a sexually active non-contracepting, and non-lactating women 15 to 49-year-old" [1]. According to the available data, 60 to 80 million couples worldwide suffer from infertility [2], but by employing various methodological approaches and different operational definitions in estimating the prevalence, research studies produced inconsistent data that make it difficult to predict the accurate global trends and estimates [3, 4]. Nevertheless, the magnitude of the problem calls for comprehensive and long-term public health interventions mainly as the majority of the causing risk factors are preventable [5, 6]. Some of the underlying causes of primary infertility have been suggested to include environmental toxins, smoking, obesity, and psychological stress [7, 8]. In addition, sociodemographic factors like age, race, education level, marital age, type of family, employment and socioeconomic status have been also evaluated for causal relationship [9, 10]. Physical activity, as an essential lifestyle modification component, has been thoroughly investigated against female infertility, but controversial research suggestions were concluded in this regard [11–13].

Health benefits has long been linked to the concept of practicing regular and adequate levels of physical activity [14, 15]. The term "physical activity", which demonstrates the skeletal muscle movements of the body that result in energy exertion, should not be confused with "exercise" that represents any planned, structured, repetitive and purposeful body movement [16]. Components of physical activity varies from leisure time physical activity, transportation, occupational and household chores to activities done during sports, games and planned exercises [17]. Globally, one from each four adults is insufficiently physically active [16], with about third of the adult population do not meet the recommended level proposed by public heath guidelines [18]. Although available data suggest that high income countries are practicing less occupational physical activities and more leisure-time among adults through time, research findings and surveillance of physical activity in most low-income countries remain scarce [18]. Gaza strip, Palestine is one of the lower middle-income countries [19], that have limited and controversial data in this regard [20, 21].

A relationship between physical activity and females' reproductive hormonal axis was found to have either beneficial or detrimental effect according to the pattern, frequency and intensity practiced [22]. While an optimal level of regular exercise is essential to maintain substantial health benefits, females practicing prolonged vigorous activities are usually exposed to various reproductive health risks [23]. Results revealed that female athletes who are exposed to intense exercise training are more prone to primary or secondary amenorrhea [24–26]. In addition, young females who practice intense exercise at young age are more likely to suffer from delayed menarche [27]. As such, the product of subtracting the energy expenditure from the dietary energy intake is the total available energy. This energy is usually required to

maintain the pulsatile pattern of the luteinizing hormone in females, which is critically important for the ovarian functionality. Any prolonged disruption in the available energy may result in suppressing the pulsatile pattern of the hormone and subsequently lead to inadequate hormonal axis required for normal menstrual cycles [28]. Furthermore, increased levels of β-endorphins and catechol estrogens and decreased levels of luteinizing hormone, follicular stimulating hormone, prolactin, 17β-estradiol and progesterone during periods of strenuous exercise may critically and reversibly induce hypothalamic anovulation [29]. On the other hand, controversial results were concluded regarding the role of various types, intensities and frequencies of physical activities in inducing infertility among females desiring pregnancy. Detrimental effect of vigorous activity has been hypothesized [30, 31], while on other occasions vigorous activity was suggested to lower the relative risk of ovulatory infertility [32], when some authors declared no relationship between the all patterns of physical activities and infertility [33].

For all what has been mentioned earlier, and for the scarcity of research related to this critical public health issue in Gaza Strip, Palestine, we conducted this study to explore the existence of potential relationship between different types, intensities and frequencies of physical activity and primary infertility among females. We hypothesized that extremes of physical activities (low or vigorous) are associated with primary infertility among females living in Gaza Strip.

## Materials and methods

### Study design and procedure

The study design of this original article was based on a retrospective observational analytic case-control study that was conducted in Gaza Strip, Palestine in 2019. The practicality and feasibility of this design was typically indicated to explore various independent variables that are associated with primary infertility. By using the observational property, the researchers intended to examine the relationship between multiple exposures against a single outcome of interest in a study population with the same study base. Adding to that, analytic methods were used to estimate the strength of associations between infertility and different variables included in this study.

### Study participants and sampling

The researchers recruited 320 couples for the study. 160 cases included married, sexually active couples in the reproductive age period (18–49), who were unable to conceive despite exposure to the risk of pregnancy for a period of at least one year. They were matched residentially with fertile couples in the reproductive age period (18–49), who were recognized as having at least two successful pregnancies with no history of assistive reproductive techniques and who were not known to have clinical infertility during their lifetime. Being a case control study, it was more appropriate to use the following in calculating the sample size; a confidence level of 95% (P < 0.05), a power of 80%, a ratio of cases to controls of 1 and a percentage of exposed controls of 50% as there is limited information about the exposure among the control group. When the total population known to have primary infertility in Gaza Strip was 15,048 couples according to what has been estimated and stated by Palestinian Center Bureau of Statistics, the researchers used epi-info 7 sample size statistical calculator and had 148 subjects for each group as the required sample size. In order to compensate missing or non-responding cases, the researchers increased the number of cases to 160 and accordingly increased the controls to 160 to have a total of 320 couples as a sample size for the study.

Multistage sampling technique was used to select the calculated sample of cases (Fig 1). To define the sample frame, five fertility centers were randomly chosen. The population frame

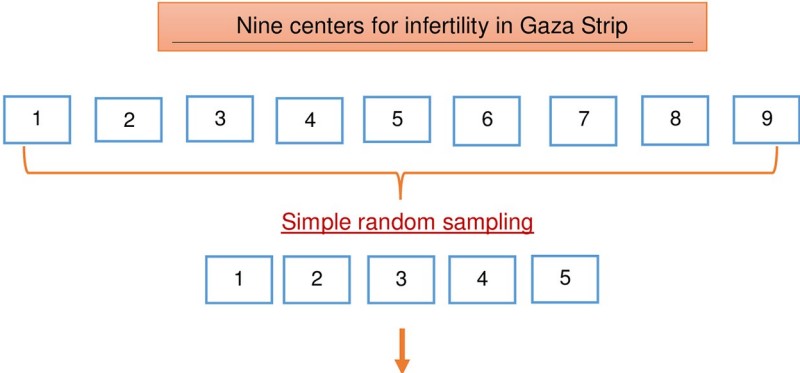

Nine centers for infertility in Gaza Strip

| 1 | 2 | 3 | 4 | 5 | 6 | 7 | 8 | 9 |

Simple random sampling

| 1 | 2 | 3 | 4 | 5 |

Cluster the lists of females registered for infertility management from January 2016 till December 2018 in each center according to the place of residency in five governorates

| Number of cases / cluster /centers | North Gaza | Gaza | Middle area | Khan Younis | Rafah | Total |
|---|---|---|---|---|---|---|
| Center 1 | 6 | 11 | 5 | 6 | 4 | 32 |
| Center 2 | 6 | 11 | 5 | 6 | 4 | 32 |
| Center 3 | 6 | 11 | 5 | 6 | 4 | 32 |
| Center 4 | 6 | 11 | 5 | 6 | 4 | 32 |
| Center 5 | 6 | 11 | 5 | 6 | 4 | 32 |
| Total | 30 | 55 | 25 | 30 | 20 | 160 |

Selecting every 4th couple through systematic stratified sampling

| Governorate | Percentage of female distribution | Distribution of cases | Number of cases / cluster / centers |
|---|---|---|---|
| North Gaza | 19% | 30 | 6 x 5 |
| Gaza | 34% | 55 | 11 x 5 |
| Middle area | 15% | 25 | 5 x 5 |
| Khan Younis | 20% | 30 | 6 x 5 |
| Rafah | 12% | 20 | 4 x 5 |
| Total | 100% | 160 | 160 |

**Fig 1. Flowchart for sampling technique and study population selection.**

was defined as the lists of couples who had registered in these fertility centers seeking medical advice for primary infertility from January 2016 till December 2018. In each center, the patients who registered in the aforementioned period were classified into sub clusters according to their residency. Finally, a 4th couple was chosen from each sub cluster (North Gaza, Gaza, Middle Area, Khan Younis and Rafah) through a systematic stratified sampling technique. Since the breakdown of total female population in reproductive age according to governorate are 88,042 in North Gaza, 155,385 in Gaza, 66,858 in Middle Area, 86,260 in Khan Younis and 55,630 in Rafah, that represent 19%, 34%, 15%, 20% and 12% respectively, the number of patients extracted systematically for each sub cluster were governed by this distribution [34]. On the other hand, the researchers sought couples of the control group from governmental primary health care clinics who approached for maternal and child care services and who were congruent with the residency of the corresponding cases.

## Study instrument

Upon reviewing literature for primary infertility in particular and reproductive health in general, a face to face interviewed questionnaire was self-constructed and developed (S1 File), typically to achieve the intended objectives and formalized to be pertinent to the environmental and social characteristics of the study setting. For validation, ten experts, including gynaecologists, epidemiologists, public health specialists and statisticians, participated in reviewing the instrument and further modification was applied upon their comments.

The questionnaire has been extended with the short form of International Physical Activity Questionnaire (IPAQ) to collect information and data about three levels of physical activity; walking, moderately intense and vigorously intense physical activity; in addition to the period spent sedentarily by examined subjects [35]. Each activity was described clearly to each participant along with the administration of show cards that contain pictures about all possible and relevant types of physical activities that could be practiced by the population in our study context [36]. Examples for moderate intensity activities were provided like, cycling, jogging, drawing water, gardening, walking with load on head and many other examples that accelerate the breathe quite more than normal, while vigorous intensity physical activities are those that make the breathe much harder than normal like, sawing hardwood, digging, shoveling sand, grinding with pestle . . .etc. Information collected for each type comprise the frequency of doing such activity in the last week and duration in minutes spent in one of these days. Responses were collected in separate forms and data was analyzed according to the recommended guidelines developed and provided by the WHO.

## Data collection

Data collectors deployed to collect information were chosen as having high experience and affinity with the data collection procedure and with the study population as well. They were provided with sufficient background knowledge about the study and were trained on the research instrument to guarantee standardization, minimize inter-observer variation and eventually assure reliability of the study. A comprehensive training was delivered on the content of the instrument, the way questions are provided, the type of terms and digits used in recording and the technique to compare responses with medical reports and records.

As a final step, a cognitive qualitative testing of the questionnaire was performed through an iterative pilot work on 30 members from the selected sample, after which the questions' format were optimized to be fully comprehended by the respondents and the instrument was explored to full achieve the purpose of the study.

## Ethical aspects

In order to launch this study, an academic approval from the School of Public Health at Al-Quds University was obtained after submitting the study proposal to the research committee for discussion. Subsequently, an ethical approval was obtained from the Palestinian Health Research Council in Gaza Strip (Helsinki Committee, approval number PHRC/HC/548/19). In the perspective of commitment to research ethics, the researchers were committed to provide an informed consent along with each questionnaire and guaranteed that each participant was fully aware and fully acquainted with each section of the attached consent form, emphasizing on their right to withdraw participation at any time. The consent explained the aim of the study along with clarification about voluntary participation with making sure that confidentiality is highly implemented. Additionally, an administrative approval was acquired from the director of Ministry of Health, as well as the specialists running the fertility centers for the purpose of having access to the institutions' database.

## Statistical analysis

Data entry was performed using IBM SPSS Statistics 22 and quality of the data was verified by refilling invalid questions through phone calls paid to the related subjects. Upon completing the process, 5% of the data was re-filled to ensure high level of procedure accuracy. Before analysis, the data was coded, where required, and was cleaned for any errors or unlogic values.

IPAQ produces two forms of outputs, one is categorical (low activity level, moderate activity level or high activity level) and the other is continuous variable (METs minutes a week). METs (Metabolic Equivalent) are typically used to express the intensity of physical activity. It is a ratio between work metabolic rate, which is the energy expended during carrying out physical work, to the standard resting metabolic rate, which is the energy expended by the body during rest period. One MET is defined as the energy expended during sitting quietly, which is equivalent to 1 kcal/kg/hour [37]. It has been estimated that moderate intensity physical activity induces four times caloric consumption compared to sitting quietly, while vigorous activity is eight times higher. The average MET value for each type of physical activity was calculated through the Ainsworth et al. Compendium (2000) of physical activity [38], where the following equations were used for analysis:

- Walking MET-minutes/week = 3.3 * walking minutes * walking days

- Moderate MET-minutes/week = 4.0*moderate-intensity activity minutes*moderate days

- Vigorous MET-minutes/week = 8.0 *vigorous-intensity activity minutes * vigorous days

- Total physical activity MET-minutes/week = Walking + Moderate + Vigorous MET-minutes/week scores

For the categorical variables, the pattern of activity was classified according to specific criteria deployed to each category. Moderate physical activity was defined when an individual practice at least 20 minutes a day of vigorous-intensity activity for 3 or more days in a typical week or 30 minutes a day of moderate-intensity activity and/or walking for 5 or more days in a typical week or total physical activity of at least 600 MET-minutes/week of any combination of walking, moderate-intensity or vigorous-intensity activities for 5 or more days in a typical week. High category is defined when an individual achieves at least 1500 MET-minutes/week of total physical activity accompanied with vigorous-intensity activity on at least 3 days in a typical week or total physical activity of at least 3000 MET-minutes/week of any combination of walking, moderate-intensity or vigorous-intensity activities for 7 or more days in a typical

week. The lowest level of physical activity is when an individual does not meet any of the above-mentioned criteria.

For statistical analysis, descriptive procedure was deployed to provide the distribution of variables in the form of central tendency (mean, median) and measures of dispersion (standard deviation, interquartile range). Crude odds ratio was calculated for each independent variable against the fertility status of both through cross tabulation chi square analysis. Nonparametric independent analysis (Mann Whitney U test) was used for ordinal not normally distributed data of metabolic equivalent and kilocalories outputs. Finally, to predict the estimated risk of each independent variable and to adjust confounding effect, an adjusted odds ratio was obtained through binary logistic regression involving independent variables that had shown statistically significant relationship with primary infertility.

## Results

A total of 320 females with a mean age of 30.2 ± 5.5 years and a median age of 30 years participated in this study. Refugee status was noticed among 70% of the infertile females, compared to 58.8% of the fertile females and almost the same frequency in both groups lived inside camps (31.2% and 32.5% respectively). Geographic characteristics revealed that 55 (34.4%) of the infertile women live in suburbs, compared to 38 (23.8%) of their counterpart, while 105 (65.6%) and 122 (76.3%) respectively live downtown with a statistically significant association. Years of education in females, their employment status and their husbands' employment status did not exhibit a significant difference between the two groups. Inquiring about average monthly income showed that 79.4% of infertile females compared to 73.8% of fertile females earn less than 440 US dollars, 10% compared to 20.6% earn 441–720 respectively and 10.6% compared to 5.6% earn more than 720 US dollars respectively. Marital age beyond 28 years was more among infertile females (17.5%) than their counterparts (2.5%), where a positive significant association was also apparent in the mean marital age (22.79 ± 6.2, 20.61 ± 3.4, $P < 0.001$). Regarding age of menarche among participants, 87 (54.4%) infertile females compared to 64 (40%) fertile females reported an age of less than 14 years for the first menstrual cycle, forming a positive significant relationship between the two groups (Table 1).

Table 2 describes the body fat distribution of participants in terms of Body Mass Index (BMI) and crude weight. BMI was calculated as the weight in kilograms divided by square the height. More than half the study population in both groups were categorized as overweight or more (58.2% of cases, 60.6% of controls). Of all infertile females, 18.8% were obese, 39.4% were overweight, 40.6% were normal and only 1.3% were underweight. The fertile group showed a distribution of 22.5%, 38.1%. 36.9% and 1.9% in each category respectively, while the mean weight was almost the same in both groups (69.5±13.2 for cases and 70.4±12.6 for controls). However, inquiry about childhood obesity was performed and a total 20 ladies reported being obese during childhood period with only 11 (6.9%) were infertile women and 9 (5.6%) were fertile. The relationship in both conditions was not significant.

Table 3 summarizes the categorical output of IPAQ analysis demonstrated in three levels of physical activities; low, moderate and high level. Positive significant association was observed between infertile females (36.9%) and fertile ones (18.8%) who were categorized as low active, where the risk was three times more likely among infertile females, OR: 3.10 (95% CI, 1.60–5.99, $P < 0.001$). This was also apparent in the time spent sedentarily, where females who spent more than 300 minutes a day sedentarily were two times more prone to primary infertility compared to those who spend less than 300 minutes a day, OR:2.27 (95% CI, 1.36–3.79, $P = 0.001$). On the other hand, fertile females (25.6%) seemed to presume high levels of activity more than infertile females (16.2%).

**Table 1. General demographic characteristics.**

| General characteristics | Category | Cases n (%) | Controls n (%) | (95% CI) | P-value |
|---|---|---|---|---|---|
| **Age** "Years" | < 25 | 35 (21.9) | 16 (10.0) | (1.33–1.18) | 0.004[†] |
| | ≥ 25 | 125 (78.1) | 144 (90.0) | | |
| | Mean ± SD | 29.8 ± 6.18 | 30.5 ± 4.76 | (1.91–0.52) | 0.261[¶] |
| **Refugee status** | Refugee | 112 (70.0) | 94 (58.8) | (1.03–2.60) | 0.036[†] |
| | Non-refugee | 48 (30.0) | 66 (41.3) | | |
| **Residency** "Area" | Suburbs | 55 (34.4) | 38 (23.8) | (1.03–2.74) | 0.036[†] |
| | Downtown | 105 (65.6) | 122 (76.3) | | |
| **Residency** "Location" | Inside camps | 50 (31.2) | 52 (32.5) | (0.59–1.51) | 0.810[†] |
| | Outside camps | 110 (68.8) | 108 (67.5) | | |
| **Years of education** | <10 | 7 (4.4) | 10 (6.3) | (0.24–1.80) | 0.409[†] |
| | 10–12 | 59 (36.9) | 62 (38.8) | (0.56–1.41) | 0.622[†] |
| | ≥ 13 | 94 (58.8) | 88 (55.0) | Ref. | |
| **Employment status** | Employed | 23 (14.4) | 31 (19.4) | (0.39–1.26) | 0.232[†] |
| | Unemployed | 137 (85.6) | 129 (80.6) | | |
| **Employment status of husband** | Employed | 120 (75.0) | 110(68.8) | (0.90–1.53) | 0.214[†] |
| | Unemployed | 40 (25.0) | 50 (31.3) | | |
| **Average monthly income** "USD/month" | ≤ 440 | 127 (79.4) | 118 (73.8) | (0.24–1.33) | 0.188[†] |
| | 441–720 | 16 (10.0) | 33 (20.6) | (0.09–0.70) | 0.007[†] |
| | > 720 | 17 (10.6) | 9 (5.6) | Ref. | |
| **Marital age** "Years" | ≥29 | 28 (17.5) | 4 (2.5) | (2.51–26.91) | <0.001[†] |
| | 18–28 | 109 (68.1) | 129 (80.6) | (0.52–1.78) | 0.908[†] |
| | <18 | 23 (14.4) | 27 (16.9) | Ref. | |
| | Mean ± SD | 22.79 ± 6.19 | 20.61 ± 3.42 | (1.06–3.27) | <0.001[¶] |
| **Age of menarche** "Years" | < 14 | 87 (54.4) | 64 (40.0) | (1.15–2.78) | 0.010[†] |
| | ≥ 14 | 73 (45.6) | 96 (60.0) | | |
| | Mean ± SD | 13.42 ± 1.35 | 13.73 ± 1.29 | (0.60–0.02) | 0.035[¶] |

[†] Pearson Chi-Square.;

[¶] Independent t test; Significant level at < 0.05.

SD = Standard Deviation; Ref. = Reference is used to breakdown more than 2x2 contingency table.

Table 4 demonstrates the second output of IPAQ analysis that shows the intensity of physical activity presumed by participants. It illustrates the distribution and relationship between three levels of metabolic equivalents exerted through walking, moderate-intense and vigorous-intense physical activity per week among infertile and fertile females. The amount of energy expended through walking by 50% of the infertile females in a week (IQR: 396, 198–594) was significantly lower than the energy expended by 50% of the fertile females (IQR: 388, 107–495). Moderate-intensity physical activity did not appear to differ much between the two groups, but more fertile females (IQR: 480, 0–480) were noticed practicing vigorous-intensity physical activity with higher metabolic rate and energy expenditure than infertile females (IQR: 720, 0–720).

Evaluating energy expenditure in kilocalories in relation to participants body weight is shown in Table 5. By measuring the number of kilocalories per week as an energy expenditure index in relation to weight, infertile females (IQR: 578, 792–214) scored significant higher range for walking than fertile females (IQR: 393, 543–150). No evidence suggested difference among those practicing moderate-intense activities, while again a significant difference

**Table 2. Description of body fat in terms of BMI and weight in kilograms.**

| Fat distribution variables: | | Cases | | Controls | | Total (%) | P-value |
|---|---|---|---|---|---|---|---|
| | | No | % | No | % | | |
| BMI | Obese | 30 | 18.8 | 36 | 22.5 | 66 (20.6) | 0.674[†] |
| | Overweight | 63 | 39.4 | 61 | 38.1 | 124 (38.8) | |
| | Normal | 65 | 40.6 | 59 | 36.9 | 124 (38.8) | |
| | Underweight | 2 | 1.3 | 4 | 2.5 | 6 (1.9) | |
| | Mean | 26.27 | | 26.77 | | 26.52 | 0.353[¶] |
| | SD | 4.94 | | 4.73 | | 4.83 | |
| Weight | Mean | 69.52 | | 70.39 | | 69.96 | 0.545[¶] |
| | SD | 13.17 | | 12.63 | | 12.89 | |
| Childhood obesity | Yes | 11 | 6.9 | 9 | 5.6 | 20 (6.3) | 0.644[†] |
| | No | 149 | 93.1 | 151 | 94.4 | 300 (93.8) | |

[†] Pearson Chi-Square;

[¶] Independent t test; Significant level at $< 0.05$;

BMI = Body Mass Index = weight in kg divided by square the height; SD = Standard Deviation.

**Table 3. Categorical output of IPAQ.**

| Physical Activity categorical output: | | Cases | | Controls | | Total (%) | 95% CI | P-value |
|---|---|---|---|---|---|---|---|---|
| | | No | % | No | % | | | |
| Categorical Scores | Low | 59 | 36.9 | 30 | 18.8 | 89 (27.8) | 3.10 (1.60–5.99) | <0.001[†] |
| | Moderate | 75 | 46.9 | 89 | 55.6 | 164 (51.3) | 1.33 (0.74–2.37) | 0.335[†] |
| | High | 26 | 16.2 | 41 | 25.6 | 67 (20.9) | Ref. | |
| Sedentary time min/day | > 300 | 55 | 34.4 | 30 | 18.8 | 85 (26.6) | 2.27 (1.36–3.79) | 0.001[†] |
| | ≤ 300 | 105 | 58.1 | 130 | 76.3 | 215 (67.2) | | |
| | Mean | 265 | | 127 | | 250 | 28.2 (2.1–54.2) | 0.034[¶] |
| | SD | 236 | | 109 | | 119 | | |

[†] Pearson Chi-Square;

[¶] Independent t test; Significant level at $< 0.05$;

Ref. = Reference is used to breakdown more than 2x2 contingency table.

**Table 4. Association of metabolic equivalent expenditure among study participants.**

| Physical Activity continuous output: | Cases | | Controls | | P-value |
|---|---|---|---|---|---|
| | Median | IQR[¶] | Median | IQR[¶] | |
| Walking MET-min/w | 396 | 396 | 264 | 388 | 0.004[†] |
| Moderate MET-min/w | 480 | 960 | 360 | 1120 | 0.725[†] |
| Vigorous MET-min/w | 0 | 480 | 180 | 720 | 0.010[†] |
| Total MET-min/w | 1302 | 1136 | 1245 | 1797 | 0.536[†] |

MET-min/w = Weekly energy expenditure in metabolic equivalents per week.

[¶] Interquartile Range;

[†] P-value of Mann Whitney U test.

**Table 5. Association of kilocalories energy expenditure among study participants.**

| Physical Activity continuous output: | Cases | | Controls | | P-value |
|---|---|---|---|---|---|
| | Median | IQR¶ | Median | IQR¶ | |
| Walking Kcal/w | 452 | 578 | 299 | 393 | 0.011† |
| Moderate Kcal/w | 549 | 567 | 442 | 1300 | 0.774† |
| Vigorous Kcal/w | 0 | 564 | 176 | 864 | 0.011† |
| Total Kcal/w | 1454 | 1562 | 1484 | 2473 | 0.513† |

Kcal/w: Weekly energy expenditure in kilocalories = MET*(weight kg/60).

¶ Interquartile Range;

† P-value of Mann Whitney U test.

**Table 6. Adjusted odds ratios of having infertility at different levels of physical activities by binary logistic regression.**

| General characteristics | Category | Adjusted OR | 95% CI | P-value |
|---|---|---|---|---|
| **Physical activity** | Low | 3.203 | 1.553–6.602 | 0.002* |
| | Moderate | 1.414 | 0.741–2.698 | 0.294 |
| | High | 1.00 | | |
| **Age** "Years" | | 0.928 | 0.880–0.978 | 0.006* |
| **Refugee status** | Refugee | 2.166 | 1.271–3.691 | 0.036* |
| | Non-refugee | 1.00 | | |
| **Average monthly income** "USD/month" | > 720 | 1.682 | 0.651–4.342 | 0.283 |
| | 441–720 | 0.499 | 0.241–1.033 | 0.061 |
| | ≤ 440 | 1.00 | | |
| **Marital age** "Years" | ≥29 | 20.731 | 5.377–79.935 | <0.001** |
| | 18–28 | 0.944 | 0.482–1.850 | 0.868 |
| | <18 | 1.00 | | |
| **Age of menarche** "Years" | < 14 | 1.820 | 1.111–2.980 | 0.017 |
| | ≥ 14 | 1.00 | | |

* Significant level at P < 0.05,

** Highly significant level at P < 0.001.

appeared between those who perform vigorous activities, $P = 0.011$. The total kilocalories exerted by both groups did not reach a significant association.

After adjusting for age, refugee status, marital age, age of menarche and average monthly income (Table 6), women with low activity were found as three times more likely among the infertile group, $P = 0.002$. Infertile females were more likely to be among the refugees group (OR: 2.17, 95% CI, 1.27–3.69), to have a marital age beyond 29 years (OR: 20.73, 95% CI, 5.38–79.93) and to have a menarche age of less than 14 years (OR: 1.82, 95% CI, 1.11–2.98).

## Discussion

This study presents evidence on the relationship between different types, intensities and frequencies of physical activity and primary infertility in Gaza Strip, Palestine. The main findings indicated that low physical activity and sedentary lifestyle behaviour in females were significantly associated with their infertility status. There was also a positive association between primary infertility and the marital age above 28 years, age of menarche below 14 years, living in suburbs and having a refugee status.

The study concluded that no significant difference was detected in relation to weight or BMI between infertile and fertile females, although more than half the study population were noticed to have a BMI above the normal recommendations stated by the WHO (18.5–24.9). Nevertheless, the study showed that more infertile females are low active OR: 3.10 (95% CI, 1.60–5.99, P < 0.001), while more females with normal fertility status perform moderate and vigorous activities. Upon adjustment for age, refugee status, marital age, age of menarche and monetary status, the risk remained almost the same showing more than three times the risk, OR: 3.203 (95% CI, 1.55–6.60, $P$ = 0.002). Likely, a study that was performed in northern China suggested that the incidence of infertility among low physically active females was four times more than that among females performing regular moderate physical activities and two times more than that of heavy vigorous exercises [39]. Regular moderate physical activity has been proven to be beneficial for maintaining and improving ovarian reserve mainly among overweight and obese women [40]. Furthermore, insulin resistance and compulsory hyperinsulinemia accompanied with low active obese females may also induce premature differentiation of granulosa cells in the small follicles caused by early response to luteinizing hormones and eventually causing anovulation [41].

In this study, sedentary time was calculated in minutes per day and included the time spent sitting at home, work, during a class or doing a course, visiting friends, reading books or watching television. Results revealed that infertile women spend more time sitting (Mean = 274.6 ± 126.5 min/d) than fertile women (Mean = 225.3 ± 102.7 min/d), where females spending more than 5 hours a day sedentarily had more than twice the risk of being infertile (OR: 2.27, 95% CI 1.36–3.79). Congruent with our results, a recent case control study that was conducted on 159 infertile and 143 fertile men and women in France, provided that sedentary behaviour of women for more than 5 hours a day was associated with more than three times the risk (OR: 3.61, 95% CI, 1.58–8.25) [42]. It has been evidenced that aerobic exercise offers a remarkable effect on the follicular phase of ovulation and enhances the development of graafian follicles mainly among women suffering from Polycystic Ovary Syndrome (PCOs) [43], while sedentary life may expose females to increase risk of having PCOs [44]. In the same respect, a large cohort prospective study conducted in the USA and examined 26,125 pregnant females and 830 infertile females known to have ovulatory causes, revealed that sedentary lifestyle and overweight held a population attributable risk for ovulatory infertility of 25% (95% CI, 20–31%) [32].

It was always believed that the regularity of the female's reproductive hormonal axis is negatively proportionated with highly intense physical activity [45]. Others thought that women practicing vigorous activity less than one hour a day will not be at risk [46], while those who consume inadequate energy producing diet along with exercising intensely are more prone to infertility [47]. Our results showed that fertile females seemed to practice more vigorous activity, where the median energy expended in this regard was zero among infertile females compared to 180 MET-min/w for those who enjoy normal fertility status, $p$ = 0.010. Furthermore, measuring energy expended by kilocalories lost in relation to weight, also revealed a negative statistically significant association between the two groups (IQR: 564 for cases, 864 for controls, $P$ = 0.011). Supporting to these findings, Rich-Edwards et al. concluded that the relative risk of ovulatory infertility decreases with each hour of vigorous activity per week by 7% and decreased to 5% after adjustment for body mass index [32]. Moreover, a metanalysis study that was performed and published recently showed that the physical activity pattern performed by women desiring pregnancy did not affect the rate of miscarriage [12], but physical activity before IVF cycles was associated with increase rate of clinical pregnancy and live births [48].

Our study has some limitations stemming from the fact that the cases were selected from women who were seeking medical advice at fertility centers. These infertile females were

economically well-off to afford seeking private reproductive health services, while those who were unable to afford the relatively expensive assisted reproductive procedures were unintentionally ignored. Furthermore, being a retrospective study, recall biases were particularly expected especially when inquiring about events that occurred in the past, especially information related to daily routines. To obtain accurate and reliable information, the participants were guided with show cards that illustrate different types and intensities of physical activities recommended and provided by the WHO guidelines for STEP wise questionnaire and precise detailed information about the frequency per day and per week were acquired. Moreover, information was verified through a reliable third-party person or with other trustworthy sources, e.g. close relative, whenever possible.

In conclusion, the study suggested that sedentary lifestyle and low activity may contribute to the risk of primary infertility among females living in Gaza Strip. The study also supported the preventive effect of vigorous activity which was congruent with some but not many research findings. It is worth to mention that the study was based on a retrospective descriptive design. These designs usually do not achieve causal-link relationships and conclusions. So, these findings may propose the need for further prospective interventional studies for more scientifically-evidenced recommendations. Furthermore, all proposed arguments in this regard, may offer the need for a unified weighting and scoring technique. This will open the gate for further related hypothesized inquiries and basically will allow comparability among countries in order to be able to speculate public health measures, given that lifestyle modification is one of the most preventable and cost-effective public health measures that has high impact on the long term.

## Supporting information

**S1 File. Primary infertility and physical activity questionnaire.**
(DOCX)

**S2 File. Readme document for the study dataset.**
(DOCX)

**S3 File.**
(SAV)

**S1 Dataset. The study dataset presented on IBM SPSS Statistics 22.**
(SAV)

## Acknowledgments

We would like to thank all females who sacrificed their time and effort in enriching this research study with valuable information, also gratitude is extended to Al-Quds University and the fertility centers in Gaza Strip for support and immense efforts.

## Author Contributions

**Conceptualization:** Amal Dhair.

**Data curation:** Amal Dhair.

**Formal analysis:** Amal Dhair, Yehia Abed.

**Investigation:** Amal Dhair.

**Methodology:** Amal Dhair.

**Project administration:** Yehia Abed.

**Resources:** Amal Dhair.

**Supervision:** Yehia Abed.

**Validation:** Amal Dhair, Yehia Abed.

**Visualization:** Yehia Abed.

**Writing – original draft:** Amal Dhair.

**Writing – review & editing:** Amal Dhair, Yehia Abed.

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
