## [Decision Letter · Decision Letter 0]

9 Jul 2020

PONE-D-20-17392

The association of types, intensities and frequencies of physical activity with primary infertility among females in Gaza Strip, Palestine: A case-control study

PLOS ONE

Dear Dr. Dhair,

Thank you for submitting your manuscript to PLOS ONE. After careful consideration, we feel that it has merit but does not fully meet PLOS ONE’s publication criteria as it currently stands. Therefore, we invite you to submit a revised version of the manuscript that addresses the points raised during the review process.

SPECIFIC ACADEMIC EDITOR COMMENTS: Two expert reviewers in the field handled your manuscript. We thank them for their time and efforts. Although interest was found in your study, several major concerns arose during review. These concerns include the need to state a directional hypothesis; questions about the experimental design; background information needs to be provided about this cohort of women, including the need to detail BMI, anthropometric information, and any data on pregnancy biometrics; and there are comments about the phrasing of the conclusions statement. All of the reviewers' comments must be addressed in the revised manuscript.

We look forward to receiving your revised manuscript.

Kind regards,

Frank T. Spradley

Academic Editor

PLOS ONE

2. Please address the following:

- Please include additional information regarding the survey or questionnaire used in the study and ensure that you have provided sufficient details that others could replicate the analyses. For instance, if you developed a questionnaire as part of this study and it is not under a copyright more restrictive than CC-BY, please include a copy, in both the original language and English, as Supporting Information.

- Please state how many participants were included in the pilot testing of this tool.

- Please provide further details of the experts employed to develop this questionnaire. For example, their levels of expertise, ages and whether or not they provided written, informed consent.

Reviewers' comments:

Reviewer's Responses to Questions

**Comments to the Author**

1. Is the manuscript technically sound, and do the data support the conclusions?

Reviewer #1: Yes

Reviewer #2: Yes

2. Has the statistical analysis been performed appropriately and rigorously? 

Reviewer #1: Yes

Reviewer #2: I Don't Know

3. Have the authors made all data underlying the findings in their manuscript fully available?

Reviewer #1: Yes

Reviewer #2: Yes

4. Is the manuscript presented in an intelligible fashion and written in standard English?

Reviewer #1: No

Reviewer #2: Yes

5. Review Comments to the Author

Reviewer #1: This study is interesting and important to know for several readers.

However, this need to be revised several points.

1) What is this study's hypothesis? Autgors need to write this in the Introduction.

2) The information of the participant’s selection is not clear. So, please add this information as a Flow chart in the results.

Reviewer #2: Manuscript Number: PONE-D-20-17392

This study reports that low levels of physical activity and sedentary lifestyle may be linked to female fertility status. This study sought to address an interesting topic that is increasingly being addressed, although there are no very large cohorts.

It is a well-written article, pleasant to read. the methodology includes a control group, which is a benefit. The authors have considered the women socio-economic condition. This study addresses a population of a region rarely studied in the field of fertility.

Main comments:

- The authors did not report any information about BMI or anthropometric status. These data are essential when dealing with physical activity or sedentary

- The authors do not provide information on previous pregnancies for the infertile women and the ages of children of fertile women. Have they had miscarriages? Previous children?

- The conclusion is too peremptory: the study shows an association between women's physical activity and fertility, but not a causal link. This would require a prospective interventional study. The authors could modulate their conclusion accordingly.

Minor comments:

- The authors report no information about male partner

- Line 132: why did the inclusion criteria for fertile women require two pregnancies? One pregnancy is enough to “label fertility”. Would it have been more appropriate to use the time to pregnancy and the date of the last pregnancy?

6. PLOS authors have the option to publish the peer review history of their article (what does this mean?). If published, this will include your full peer review and any attached files.

Reviewer #1: No

Reviewer #2: No

---

## [Author Response · Author response to Decision Letter 0]

17 Sep 2020

Dear academic editor and reviewers,

Please, let us thank you for your time and effort in editing and reviewing our manuscript and please, kindly find replies for your comments:

- Please include additional information regarding the survey or questionnaire used in the study and ensure that you have provided sufficient details that others could replicate the analyses. For instance, if you developed a questionnaire as part of this study and it is not under a copyright more restrictive than CC-BY, please include a copy, in both the original language and English, as Supporting Information.

Questionnaire is added as a supporting information. Also, the study dataset was previously uploaded in the supporting information.

- Please state how many participants were included in the pilot testing of this tool.

Added in line 195

- Please provide further details of the experts employed to develop this questionnaire. For example, their levels of expertise, ages and whether or not they provided written, informed consent.

The questionnaire was self-developed except for the part that includes physical examination part in which it was adopted from the Short Form of International Physical Activity Questionnaire of the World Health Organization template. The whole tool is uploaded as a supporting information. 

3. Please include captions for your Supporting Information files at the end of your manuscript, and update any in-text citations to match accordingly.

Reviewer #1: 

1) What is this study's hypothesis? Authors need to write this in the Introduction.

The study hypothesis has been added to the introduction (lines 122 - 126). Thanks for your comments.

2) The information of the participant’s selection is not clear. So, please add this information as a Flow chart in the results.

A flow chart has been added to the methodology section at the study participants and sampling part as Fig 1. Thank you for your enriching comments.

Reviewer #2: Manuscript Number: PONE-D-20-17392

Main comments:

- The authors did not report any information about BMI or anthropometric status. These data are essential when dealing with physical activity or sedentary

Yes, you are right. It is crucial to explore anthropometric measures when we are dealing with physical activity. We have added information related to weight, BMI and childhood obesity to the results and discussion section (lines 269 - 281). Thanks for the comment.

- The authors do not provide information on previous pregnancies for the infertile women and the ages of children of fertile women. Have they had miscarriages? Previous children?

The cases were selected upon a criterion of having primary infertility. To be primary infertile they will have a zero history of pregnancies. For fertile women, the selection criteria included those who had no previous or current bad obstetric and perinatal histories. That’s why we did not inquire about previous miscarriage. 

- The conclusion is too peremptory: the study shows an association between women's physical activity and fertility, but not a causal link. This would require a prospective interventional study. The authors could modulate their conclusion accordingly.

Yes, actually it is correct and we should not have been so definitive in stating the conclusion. Observational studies are suggestive. We have modulated the conclusion accordingly (Lines 388 – 393).

Minor comments:

- The authors report no information about male partner

Actually, we did not include the male partner in this study.

- Line 132: why did the inclusion criteria for fertile women require two pregnancies? One pregnancy is enough to “label fertility”. Would it have been more appropriate to use the time to pregnancy and the date of the last pregnancy?

We have chosen controls upon which those who had at least two successful pregnancies without assistive reproductive techniques to exclude secondary infertility. Other inclusion criteria for controls was to have no bad obstetric or perinatal history like miscarriage, still births, early perinatal deaths but this was not mentioned in the methodology. It has been added, so that the selection criteria would be clearer. Regarding time to pregnancy, we considered the available cultural norms and traditions unique to this community. Here, couples do not delay planning to pregnancy once they are married and they are always anxious and otherwise would seek medical advice immediately. Still this could be mentioned as a limitation. Thanks for your enriching comments.

Finally, I would like to thank you all for the effort and time and for enriching our research with your valuable comments.

Best regards,

---

## [Decision Letter · Decision Letter 1]

8 Oct 2020

The association of types, intensities and frequencies of physical activity with primary infertility among females in Gaza Strip, Palestine: A case-control study

PONE-D-20-17392R1

Dear Dr. Dhair,

We’re pleased to inform you that your manuscript has been judged scientifically suitable for publication and will be formally accepted for publication once it meets all outstanding technical requirements.

Kind regards,

Frank T. Spradley

Academic Editor

PLOS ONE

Reviewers' comments:

Reviewer's Responses to Questions

**Comments to the Author**

1. If the authors have adequately addressed your comments raised in a previous round of review and you feel that this manuscript is now acceptable for publication, you may indicate that here to bypass the “Comments to the Author” section, enter your conflict of interest statement in the “Confidential to Editor” section, and submit your "Accept" recommendation.

Reviewer #1: All comments have been addressed

Reviewer #2: All comments have been addressed

2. Is the manuscript technically sound, and do the data support the conclusions?

Reviewer #1: Yes

Reviewer #2: Yes

3. Has the statistical analysis been performed appropriately and rigorously? 

Reviewer #1: Yes

Reviewer #2: I Don't Know

4. Have the authors made all data underlying the findings in their manuscript fully available?

Reviewer #1: Yes

Reviewer #2: Yes

5. Is the manuscript presented in an intelligible fashion and written in standard English?

Reviewer #1: Yes

Reviewer #2: Yes

6. Review Comments to the Author

Reviewer #1: Thanks for your revision based on the reviewer's comments.

There is no any comments. So, I recommend to publish this.

Reviewer #2: This study sought to address an interesting topic that is increasingly being addressed, although there are no very large cohorts. The authors responded correctly to the comments. The article meets the conditions for publication.

7. PLOS authors have the option to publish the peer review history of their article (what does this mean?). If published, this will include your full peer review and any attached files.

Reviewer #1: No

Reviewer #2: **Yes: **Charlotte Dupont

---

## [Editor Report · Acceptance letter]

12 Oct 2020

PONE-D-20-17392R1 

The association of types, Intensities and frequencies of physical activity with primary infertility among females in Gaza Strip, Palestine: A case-control study 

Dear Dr. Dhair:

I'm pleased to inform you that your manuscript has been deemed suitable for publication in PLOS ONE. Congratulations! Your manuscript is now with our production department. 

Kind regards, 

on behalf of

Dr. Frank T. Spradley 

Academic Editor

PLOS ONE